# Sanitizers Used for Fungal Spoilage Control in Dry-Fermented Cured Meat Production

**Sarah Silva, Angélica Olivier Bernardi, Marcelo Valle Garcia** [ID]**, Thais Nunes Bisello, Larissa Borstmann and Marina Venturini Copetti *** [ID]

Postgraduate Program in Food Science and Technology, Center of Rural Sciences, Federal University of Santa Maria, Santa Maria 97105-900, Brazil
* Correspondence: marina.copetti@ufsm.br

**Abstract:** Contamination caused by fungi stands out as a significant microbiological issue in the food industry, particularly leading to premature spoilage across various food segments, including the dry-fermented meat industry. The emergence of undesired fungi on product surfaces results in substantial economic losses. Once microorganisms infiltrate the food, contamination ensues, and their subsequent proliferation can adversely impact the product's appearance, odor, flavor, and texture. This, in turn, leads to consumer rejection and negatively affects the commercial brand. Additionally, concerns persist regarding the potential presence of mycotoxins in these products. Given the detrimental effects of spoilage fungi in the food industry, practices such as thorough cleaning and sanitization become crucial to prevent contamination and subsequent premature deterioration. These measures play a pivotal role in ensuring the quality and safety of food, while also extending the shelf life of products. This review delves into the advantages, disadvantages, and factors that may influence the efficacy of commonly used sanitizers in the dry-fermented cured meat industry, including substances like sodium hypochlorite, peracetic acid, and benzalkonium chloride.

**Keywords:** sanitizers; fungal spoilage; food industry





## 1. Introduction

Dry-fermented meat products are widely consumed worldwide, and the occurrence of common molds on their surfaces is generally deemed normal. This can even be considered a quality indicator, as long as these molds do not synthesize mycotoxins or antibiotics [1]. Fungi play a pivotal role in the technological process by releasing enzymes that elevate the sensory characteristics of dry-fermented products, resulting in distinctive flavors [2]. However, a potential drawback exists, as undesirable species capable of producing mycotoxins may also develop on the product surface, posing a threat to consumer exposure to harmful compounds [1,3–6].

The capacity of fungi to thrive in acidic pH, along with their resilience to the low pH and high salt concentration found in dry-fermented meat products, promotes the growth of filamentous fungi over other microbial groups [4,7]. The richness and diversity of species existing in raw materials and the production environment of dry-fermented meat products play a significant role in shaping the mycobiota of the end product [8–10].

Species that produce ochratoxin A are especially undesirable in dry-fermented meat products. In temperate climates, the most relevant are *Penicillium nordicum* and *Penicillium verrucosum* [4,5,7–9], while in warmer climates, *Aspergillus* from Section *Circumdati* (mainly *A. westerdijkiae* and *A. ochraceus*) stands out, both in South America and Mediterranean countries [4,7,8,10]. The occurrence of both fungi and mycotoxins in these products is linked to contamination from raw materials, particularly spices, and the air in the maturation chamber, where products of different ages often mature together [2,8,10–12]. A study conducted by Almeida [11] demonstrated a two-log increase in the fungal population when comparing cured sheep ham with spices to ham produced without this raw material.

Contamination by fungi, encompassing molds and yeasts, is a critical factor contributing to losses and waste resulting from premature fungal spoilage in food. Fungi, predominantly molds, pervade the entire food production chain and can originate from various stages, including seed contamination, cultivation, harvesting, post-harvest activities, food processing, transportation, and storage [3]. This pervasive presence poses a significant risk, leading to substantial economic losses and potential health hazards for consumers. Certain fungi have the capacity to produce toxic secondary metabolites, such as mycotoxins, with adverse effects on both human and animal health [1,4]. Furthermore, the proliferation of fungi in food is associated with adverse effects on the sensory attributes of products, such as appearance, texture, and flavor properties [3,5]. These consequences not only prompt consumer rejection but also contribute to economic losses for producers.

Then, the presence of undesirable fungi in the meat facility environment can lead to significant economic losses when potential spoilage (especially mycotoxin-producing species) present in production and processing environments contaminates the fresh processed meat product. This contamination can result in the subsequent multiplication of these agents, leading to undesirable changes and final product deterioration [13]. To counteract early fungal spoilage, one strategy employed by the food industry involves the implementation of a good hygiene process that encompasses cleaning methods plus adequate sanitization [14–17]. The cleaning step is essential for decreasing the amount of organic load (soil) in the meat processing facility because these compounds can strongly reduce the sanitizers' efficacy against fungal species involved in the spoilage of dry-cured meats [17]. Therefore, the hygiene process applied in a meat processing facility is crucial to reducing the fungi contaminating the work surfaces and environment to a safe level. This, in turn, will impact the fungal load in the food, which ultimately can extend the shelf life of the product [15].

Achieving effective fungi control in a dry-fermented meat products facility relies heavily on maintaining hygiene in the production and maturation environment, encompassing air quality and work surfaces [13,15,17]. Additionally, the use of sanitizing agents with proven and suitable antifungal properties for work surfaces, air, and processing and maturation environments is crucial [13,16]. Beyond selecting the optimal active ingredient through laboratory tests, it is essential to take into account external factors such as exposure time, temperature, concentration, type of disinfectant, and the presence of organic loads, as these can influence the antifungal activity of sanitizers [17,18]. The mode of application (fumigation, liquid application) and the specific fungi targeted also play a role [3].

Emphasizing a study conducted by Bernardi et al. [15], which brought to light the significant tolerance demonstrated by fungal strains (*A. westerdijkiae*, *Penicillium polonicum*, and *Aspergillus pseudoglaucus*) isolated from dry-fermented meat products toward widely used sanitizers in the food industry, even when applied at recommended maximum dosages. The research indicated that the lowest concentration specified on the product label, closely aligned with industry hygiene practices, consistently proved ineffective against the tested spoilage species.

In an effort to elucidate the factors affecting the antifungal efficacy of sanitizers, Stefanello et al. [18] underscored the importance of an efficient cleaning step preceding the sanitization process to enhance its effectiveness. The presence of an organic load was identified as a factor that diminishes the efficacy of most sanitizers. Additionally, the duration of sanitizer contact is crucial, with a minimum of 15 min of product action recommended. Temperature variations also play a role, as peracetic acid demonstrates greater effectiveness at higher temperatures (40 °C), while benzalkonium chloride tends to exhibit optimal action at lower temperatures (10 °C).

These findings find support in the work of Silva et al. [17], who evaluated the impact on *P. verrucosum*, *P. nordicum*, and *A. westerdijkiae*—species identified as significant contributors to the spoilage of dry-fermented meat products due to their ochratoxin A production. The study noted a higher sensitivity of the studied *Penicillium* species compared to the *Aspergillus* species. Each sanitizer presents its own set of advantages and disadvantages

concerning microorganism reduction, cost, and accessibility. Furthermore, each sanitizer differs in its permitted limits. Table 1 provides a compilation of the most commonly used sanitizers with approved use in food production environments.

**Table 1.** Sanitizers, active ingredients, and concentrations indicated by the manufacturer.

| Sanitizer | Active Principle | Suggested Use Concentration |
|---|---|---|
| Benzalkonium chloride | Benzalkonium chloride | 0.3–5% |
| Sodium hypochlorite | Sodium hypochlorite, 10–12% of active chlorine | 0.1–1% |
| Peracetic acid | Peracetic acid, hydrogen peroxide, acetic acid | 0.15–3% |

Hence, the primary goal of this review is to comprehensively discuss the key aspects and features pertaining to commonly used sanitizers in the food industry: peracetic acid, sodium hypochlorite, and benzalkonium chloride. The focus is on their efficacy in controlling spoilage in dry-fermented meat, coupled with insights into the utilization of these sanitizers for managing fungal spoilage in general. This review aims to provide information on the advantages and disadvantages of each sanitizer, as well as factors influencing their effectiveness. By addressing these decisive factors, this review aims to assist the food industry in effectively managing early fungal spoilage and mitigating associated challenges.

## 2. Fungal Control through the Hygiene Process

To guarantee the production of high-quality products, predominantly free from microbiological pathogens and spoilage agents, the food industry needs to establish measurable and monitorable limits. These limits should ensure the effectiveness of procedures and the attainment of predefined objectives [19]. Both the sanitizer and the hygiene process should facilitate the production of food with an extended shelf life while ensuring the safety of consumers' health [20,21].

For the process to be effective, it is crucial to choose sanitizers that contain active ingredients proven to be effective against the target microorganisms. The concentrations applied should be adequate for fungal inactivation without unnecessary waste, adhering to microbiological recommendations set with technical criteria for sanitized surfaces, processing environments, food handlers, and equipment [13,22].

A critical aspect of the sanitization phase within an effective hygiene process is the careful selection of the sanitizer. Several factors must be taken into account, including the spectrum of action, antimicrobial or antifungal activity [23], formation of toxic by-products [24] and whether the sanitizer complies with safety and legal requirements stipulated by relevant regulatory bodies [14].

A sanitizer can only be registered and authorized for use after demonstrating its antimicrobial efficacy for the intended purposes. This verification is typically conducted through efficacy testing on the finished product and in the dilutions specified for use by the manufacturer on the label. These analyses may follow the methodology of the Association of Official Analytical Chemists (AOAC) or methods endorsed by the European Committee for Standardization (CEN) for liquid sanitizers (European Standard n 13697) [25]. For smoke sanitizers, compliance with the French Standard (NF-T-72281) [26] is essential, as it outlines the methodology for evaluating the effectiveness of smoke-generating agents. Figure 1 provides an overview of the sanitizing efficacy analysis scheme based on the modeling of efficacy tests as described by Bernardi et al. [15,23].

The selection of a sanitizer should take into consideration various factors, including the type of equipment surface and the specific location to be disinfected, the presence of residual organic load (soil), temperature, water quantity, contact time, the spectrum of action of the agent, and the residual efficacy of the product, among other considerations [14]. Unfortunately, it is a common occurrence for sanitizers to be employed in inappropriate concentrations or combined with multiple products, resulting in formulations that may compromise the antimicrobial activity of the product [22].

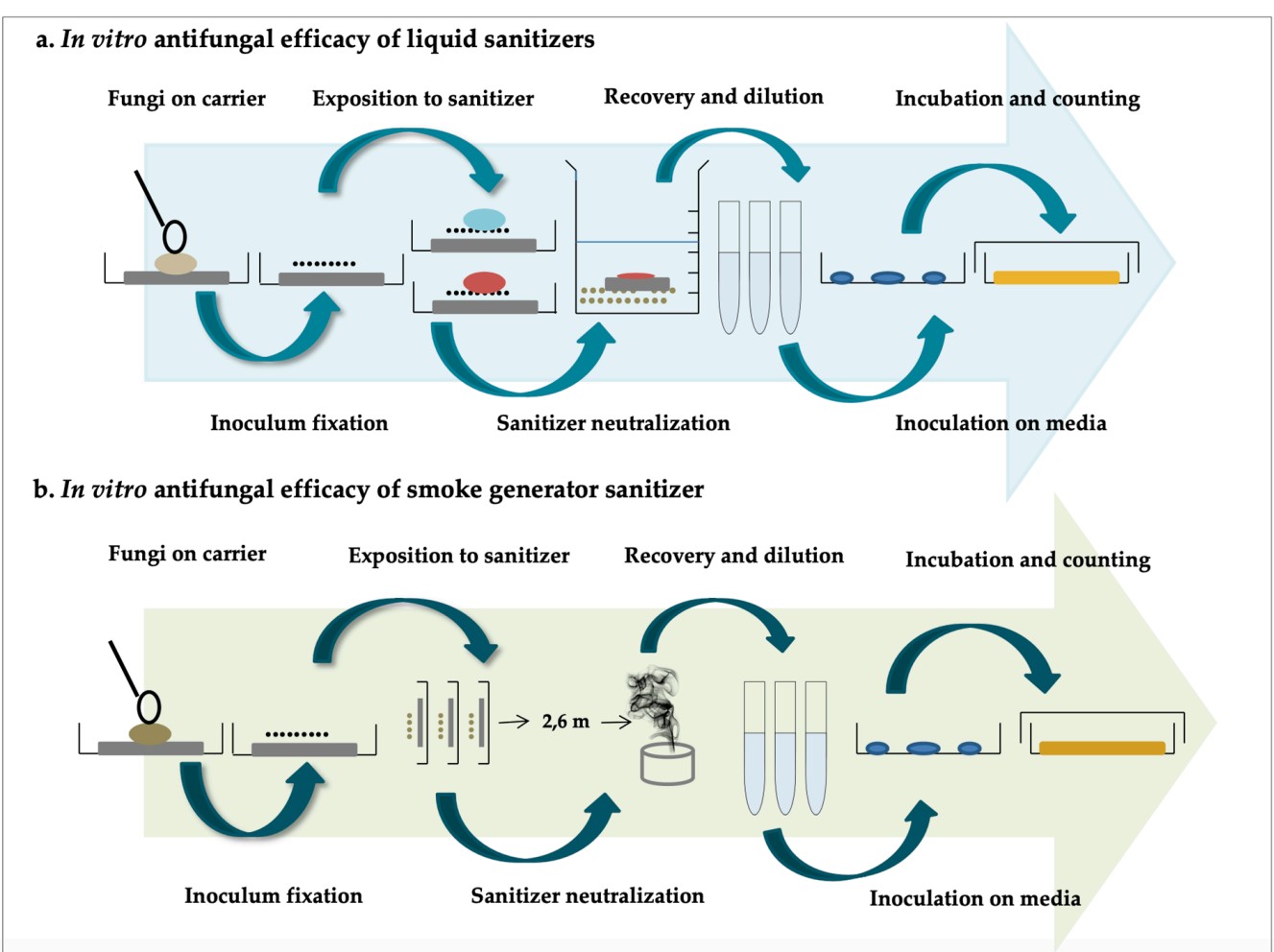

**Figure 1.** Schematic of in vitro efficacy testing for liquid (**a**) and smoke (**b**) sanitizers.

In the following section, some common sanitizers used for fungal control in the food industry for the disinfection of work surfaces, air, and production and processing environments will be discussed, summarizing their advantages and disadvantages in Table 2.

### 2.1. Sodium Hypochlorite

Sodium hypochlorite stands out as the most commonly used sanitizer, often serving as a reference in comparative analyses of sanitizers [24,25]. The mechanisms of action of sodium hypochlorite are grounded in its physicochemical properties [26]. Sodium hypochlorite (NaOCl) reacts with water to produce hypochlorous acid (HOCl) [27], also known as free chlorine, through hydrolysis (NaOCl + $H_2O$ → HOCl + $NaOH^-$). Subsequently, hypochlorous acid dissociates into the hypochlorite ion ($ClO^-$) and $H^+$ (proton) [28,29]. In solution, the hypochlorite ion ($ClO^-$), one of the active oxidizing forms, directly acts on microorganisms, rendering them inactive by inhibiting enzymatic reactions, denaturing proteins, and inactivating nucleic acids within the cells [30]. This agent serves as a chlorine source and is recognized as a potent oxidizing agent. It is extensively used for cleaning and sanitization in the food industry. Notable features of this compound include its broad spectrum of activities, highlighted by its whitening action [31] and cost-effectiveness [32–34], along with its minimal impact on the nutritional qualities of food [35].

The permissible maximum concentration of sodium hypochlorite for use on food contact surfaces typically ranges from 0.005 to 0.02% (50 to 200 parts per million) and 0.05 to 0.08% (500 to 800 parts per million) in non-food contact areas. However, efficacy tests

on non-food contact surfaces have indicated that while sodium hypochlorite effectively combats bacteria and yeasts, it is unable to achieve a 4-log inactivation of fungal spores for most tested species [16,18,36,37], including dry-meat spoilage fungal species [17]. This specified concentration is outlined in legislation for assessing the efficacy of sanitizers permitted for use in the food industry. Considering these results, sodium hypochlorite may not be an adequate choice in industries aiming for fungal inactivation, as it exhibits limited effectiveness in controlling fungal contamination [36–38].

2.1.1. Factors Influencing the Efficacy of Sodium Hypochlorite

Several factors influence the efficacy of a sanitizer, and one of the primary elements leading to a reduction in the effectiveness of sodium hypochlorite is the presence of organic loads. The presence of organic loads can generate by-products and diminish the antimicrobial action of sodium hypochlorite [39–43]. In vitro results obtained from assessing the interference of various factors on the antifungal action of sodium hypochlorite against fungi of the ochratoxin-producing species *P. nordicum*, *P. verrucosum*, and *A. westerdijkiae*, which are among the most relevant spoilage fungi in dry-fermented meat products, indicated that organic load was a key factor responsible for diminishing the antifungal action of sodium hypochlorite [17,18]. The organic load used in these in vitro assessments aimed to simulate the presence of dirt in industrial environments, and the results demonstrated an increase in the survival of all fungal species. This underscores the importance of the cleaning stage, involving the removal of organic loads in food facilities, as a crucial procedure when the objective is to reduce the fungal load [17,18,44,45].

Another crucial factor for effective antimicrobial action is the concentration of the sanitizer in use. Bernardi et al. [23] emphasize the significant importance of choosing the right concentration to achieve satisfactory results, suggesting that the highest concentration recommended by the manufacturer should be preferably used, as it often demonstrates effectiveness against fungi. In studies by Ribeiro et al. [46], sodium hypochlorite was considered more effective than peracetic acid against *Aspergillus nomius* inoculated in Brazil nuts, even though it achieved a reduction of less than 2 log CFUs when exposed for 8.5 min at a concentration of 250 ppm. Similarly, in in vitro efficacy tests, Gonçalves et al. [38] observed that concentrations between 500 and 750 ppm of sodium hypochlorite, applied for 15 min, inactivated between 2 and 2.9 log CFUs of this species. Consistent with the findings of Salomão et al. [47], who applied this sanitizer to apples inoculated with *Penicillium* spp., a higher reduction was achieved by increasing the concentration from 50 to 200 mg/L. In a study conducted by Bernardi et al. [36], higher concentrations of sodium hypochlorite were applied than typically recommended, demonstrating that this agent was effective only at concentrations of at least 5000 ppm against *Hyphopichia burtonii* and 10,000 ppm for *P. roqueforti*, *Penicillium paneum*, and *Aspergillus pseudoglaucus*, with exposures of 15 min.

The effectiveness of sodium hypochlorite is significantly influenced by the pH of the environment [42,48–50]. The antimicrobial action of these compounds is typically associated with the generation of hypochlorous acid following hydrolysis in water. The formation of hypochlorous acid tends to be higher when the pH is slightly acidic because this form predominates as the antimicrobial agent, as opposed to the hypochlorite ion, which becomes dominant at pH values above 7.0. In such alkaline conditions, sodium hypochlorite-based compounds are less effective [51]. In studies conducted by Salomão et al. [47], chlorine solutions at a pH of 6.5 resulted in reductions of more than 5 logs in *Penicillium expansum* spores in Macintosh, Empire, and Golden Supreme apples at a concentration of 200 ppm.

**Table 2.** Interference factors, advantages, and disadvantages of common sanitizers in the food industry [14,44,45,51].

| Sanitizer | Interference Factor | Advantages | Disadvantages |
|---|---|---|---|
| Belzalkonium chloride | pH. | Low toxicity. | Gram⁻ bacteria tolerance |
| | Temperature. | Ability to be formulated for specific objectives. | Unrestricted use |
| | Concentration. | Food preservative. | Emerging pollutant. |
| | Anionic detergents. | Residual action. | Toxic to many species of aquatic and terrestrial organisms. |
| Sodium hyphoclorite | Organic load | Chloride source. | Toxic to the environment. |
| | Concentration | Low cost. | Corrosion. |
| | Exposition time | Low effect on nutritional qualities. | Residual toxicity. |
| | pH. | Bleach action. | Generation of disinfection by-products. |
| Peracetic acid | Temperature | Sustainable. | Slow inactivation kinetics. |
| | Organic load | Low environmental impact. | Organic content in the effluent. |
| | | Prevent biofilms. | Microbial regeneration potential. |
| | | | Formation of acetic acid in high concentration. |
| | | | Higher cost. |
| | | | Damage to skin, eyes and respiratory tract. |

2.1.2. Disadvantages of Sodium Hypochlorite

While higher concentrations of sodium hypochlorite can enhance its antimicrobial efficacy, such concentrations are not recommended due to the potential generation of toxic compounds for the environment, corrosion in materials and equipment, the risk of explosions, and adverse effects on workers' health [41,43,52,53].

Sodium hypochlorite exhibits residual toxicity, necessitating specific precautions during its application, handling, and storage. The use of personal protective equipment is advised, along with thorough hand washing procedures [44,45]. A primary concern associated with the use of sodium hypochlorite is the reactivity of chlorine with organic loads, resulting in the generation of disinfection by-products [24,54,55]. These by-products, along with the potential negative consequences on the environment and human health [24,41–43,52,53,56–58], may lead to negative sensory effects, such as unpleasant odors and taste in fresh produce [43,59].

Additionally, stability and the potential for damage to equipment surfaces are relevant factors in the application of chemical products. Sodium hypochlorite, being a major corrosive agent on metal surfaces, including stainless steel [44], can lead to damage and, after corrosion, may result in the accumulation of organic load. This accumulation can make sanitation challenging and contribute to the development of microbial biofilm [36,45].

*2.2. Peracetic Acid*

Considered an environmentally friendly oxidant and disinfectant with low environmental impact [60,61], peracetic acid stands out as a key alternative to chlorinated

compounds [62–65]. One of its notable advantages is its minimal reactivity with proteins, effectively preventing the formation of biofilms [66].

Commercially, peracetic acid is available as a mixture containing hydrogen peroxide ($H_2O_2$) (10–40%), acetic acid (3–40%) [24,32,67,68], and water [69]. This agent, known for its environmental compatibility, decomposes into harmless derivatives (acetic acid, water, and oxygen) [44], which are swiftly metabolized by microorganisms [70]. Possessing lipid solubility properties, as highlighted by Lazado et al. [71], peracetic acid acts directly and robustly on cell membranes through hydroxyl radicals [72]. Reactive oxygen species cause damage to DNA and lipids, disrupting membranes, and blocking enzymatic and transport systems [73].

Owing to its demonstrated fungicidal and sporicidal efficacy in various applications, the use of peracetic acid as a disinfectant in the food industry has gained increased attention in recent years [74]. It is considered highly effective and is applied in different environments, including food processing, beverages, water from cooling towers, and wastewater [62]. In the minimally processed industry, which seeks sustainable alternatives to chlorine [54], peracetic acid is also applied to food contact surfaces [75]. Furthermore, it finds use in cheese and meat facilities [15], bakeries [36], and is also effective for controlling mycotoxin-producing species [38]; usually reaching high fungal inactivation when used in intermediate to high concentrations.

### 2.2.1. Factors Influencing the Efficacy of Peracetic Acid

Peracetic acid demonstrates lower susceptibility to the presence of organic loads compared to chlorine [24,76]. It is less affected by water from the process and does not lead to the formation of disinfection by-products or does so to a limited extent [24,52,69,77,78]. Additionally, it exhibits good stability, a high redox potential (1.8 eV) [79], operates efficiently across a wide pH range (1 to 8) [24,52,80], and functions effectively within a temperature range of 0 to 40 °C [17,52]. However, it comes with a higher cost compared to chlorine, and its kinetics of slower inactivation have been noted [81,82].

Over the years, research on the antifungal efficacy of peracetic acid has expanded. Studies against *Penicillium digitatum* found effectiveness at concentrations of 72 ppm for 8 min at 25 °C and 216 ppm for 30 s at 35 °C [83]. At concentrations up to 3.0%, it demonstrated reductions between 2 and 4 logs for bakery spoilage fungi [36], and for industrial purposes in thermosensitive strains (*Cladosporium cladosporioides*, *Penicillium commune*, *Penicillium polonicum*, and *Penicillium roqueforti*) [23]. In the case of Aspergillus brasiliensis (ATCC 16404), peracetic acid exhibited satisfactory action, completely inactivating the species under various tested conditions, such as 30 and 40 °C at 1% for 10 and 15 min and a 5 min exposure time at 40 °C in both conditions (with/without organic load) [18].

Observations aligned with studies by Silva et al. [17], which evaluated fungal species isolated from spoiled meat products, and Stefanello et al. [18], which, dealing with standard strains, suggested that at higher temperatures, the antifungal action of peracetic acid is enhanced. For instance, when applied at 40 °C for 40 s at a concentration of 100 mg/L, its action was significantly more pronounced than the same treatment at room temperature (20 °C), completely inhibiting *Monilinia fructicola* conidia in water [80].

### 2.2.2. Disadvantages of Peracetic Acid

Research indicates that peracetic acid reacts with amino acids, phenols, and other aromatic substances in wastewater, leading to the formation of approximately 10–30 µg/L of aldehydes [83]. In a study by Lee et al. [24], comparing minimally processed lettuce washing water with sodium hypochlorite and peracetic acid at different concentrations, it was found that peracetic acid washing generated disinfection by-products, albeit in smaller quantities compared to sodium hypochlorite. These findings contrast with studies by Cavallini et al. [84] and Araújo et al. [85], who reported limited or undetectable amounts of halogenated disinfection by-products during wastewater application. Further research

is necessary to elucidate the formation of disinfection by-products resulting from peracetic acid treatment.

The use of peracetic acid may also present other disadvantages [86], including a higher organic content in the effluent, microbial regeneration potential, reduced efficiency against viruses and parasites, unpleasant odors generated by the formation of acetic acid in high concentrations, and potential irritation to the skin, eyes, and respiratory tract [87,88]. The acetic acid released after decomposition and hydrolysis in water [89] can contribute to regeneration, resulting in an increase in organic load in wastewater or water treated with peracetic acid [80,90].

### 2.3. Benzalkonium Chloride

Benzalkonium chloride belongs to the group of quaternary ammonium compounds (QACs) [91], specifically from the first generation [36]. Quaternary ammonium compounds typically have at least one long hydrophobic alkyl chain substituent at one end and a short alkyl chain (methyl, benzyl, or ethyl benzyl) at the other end of the quaternary ammonium cation [92]. The antimicrobial activity of these compounds depends on the length of the alkyl chain, with homologous C12 being effective against yeasts and molds, C14 acting well on Gram-positive bacteria, and C16 on Gram-negative bacteria [93]. These compounds can be formulated for specific target microorganisms [94–96] and are known for their low toxicity [97–100]. In lower concentrations (0.5 to 5 mg/liter), quaternary ammonium compounds, including benzalkonium chloride, also exhibit fungistatic properties [101].

The performance of benzalkonium chloride against food spoilage fungi has shown promising results. Evaluations of its efficacy at different concentrations (0.3%, 2.5%, and 5%) against spoilage fungal species from bakery products (such as *Penicillium roqueforti*, *Penicillium paneum*, *Hyphopichia burtonii*, and *Aspergillus pseudoglaucus*) revealed its effectiveness in inactivating strains of *P. roqueforti* [36]. When exposed to different concentrations of benzalkonium chloride, fungi associated with the spoilage of dairy and meat products, including *A. westerdijkiae*, *A. pseudoglaucus*, *Penicillium commune*, *P. roqueforti*, and *P. polonicum*, exhibited varying degrees of resistance, with meat product spoilers generally showing higher resistance to this sanitizer [16]. Studies have also reported its effectiveness against aflatoxigenic fungi, with benzalkonium chloride proving effective against multiple strains of *Aspergillus* spp. [38]. Additionally, it has demonstrated good antifungal action against heat-resistant strains of *Paecilomyces variotii*, *Paecilomyces niveus,* and *Aspergillus neoglaber* [37].

### 2.3.1. Factors Influencing the Efficacy of Benzalkonium Chloride

Quaternary ammonium compounds, including benzalkonium chloride, are known for their stability over a wide temperature range. They also exhibit some detergency, resulting in higher stability in the presence of organic loads and greater activity in alkaline pH conditions [102]. The antifungal action of benzalkonium chloride against ochratoxigenic fungi, such as *P. nordicum*, *P. verrucosum*, and *A. westerdijkiae*, has shown positive results, even under different conditions of exposure time, concentration, and temperature. Studies have reported that benzalkonium chloride exhibits effective antifungal action, particularly at lower temperatures (10 and 25 °C). However, its efficacy can be negatively affected in the presence of organic loads, simulating a dirty environment [17]. Stefanello et al. [18] also noted greater efficacy of benzalkonium chloride at low temperatures (10 °C) against *A. brasiliensis* (ATCC 16404). The favorable action at low temperatures is advantageous for the food industry, allowing for cost savings as there would be no need to heat washing water. This enables the use of wells and reservoirs, especially when increasing the temperature throughout the process is neither desired nor required by legislation.

### 2.3.2. Disadvantages of Benzalkonium Chloride

Insufficient rinsing and sanitizer residue on the surface of equipment can select more tolerant populations and lead to the development of resistance in microorganisms. Com-

post waste of quaternary ammonium has been found in various types of foods, such as meat, dairy products, fruits, and nuts [103,104]. Due to their wide application, quaternary ammonium compounds have unrestricted use as active ingredients in many commercial sanitizers. As a result, it has been considered a type of emerging pollutant [105]. The presence of quaternary ammonium compounds in soil and water prevents the biodegradation of natural environments due to their toxicity to many species of aquatic and terrestrial organisms [100]. Furthermore, there is the exposure of microorganisms to sub-lethal concentrations of sanitizer [106], facilitating tolerance acquisition by different species and leading to co-resistance and cross-resistance to other antimicrobial agents such as antibiotics [107].

## 3. Conclusions

The search for effective sanitizers against food spoilage fungi remains a challenge for the food industry. Recent studies have evaluated the in vitro antifungal efficacy of various active ingredients in sanitizers permitted for use in the food industry to control fungi that are relevant as spoilage agents. However, a common finding is that most sanitizers achieve the required fungal inactivation only when tested at the highest concentration specified on the product label. In practice, the food industry tends to use lower concentrations, which are often ineffective.

Despite ongoing research, there is still a need for more information on the mechanisms of action of sanitizing agents on fungal cells, the impact of combining different agents, efficacy against fungal biofilms, and on-site testing. It's crucial to recognize that there is no ideal antifungal sanitizer. Each compound has advantages and disadvantages, and its action is influenced by various factors. There is a consensus that variability exists in the sensitivity of fungi to sanitizing agents, emphasizing the importance of using the highest concentrations recommended on the product label for a satisfactory spectrum of action, typically for at least 10 to 15 min.

Fungal control in the food industry demands careful attention and commitment from personnel involved in quality control. Developing hygiene plans that consider the correct and most common factors influencing the antifungal action of each sanitizer is essential for achieving the desired results. This involves a comprehensive approach to sanitation to ensure the production of high-quality, contamination-free food products.

**Author Contributions:** Conceptualization, S.S., A.O.B. and M.V.C.; formal analysis, S.S., T.N.B. and L.B.; resources, M.V.C.; data curation, S.S. and M.V.C.; writing—original draft preparation, S.S.; writing—review and editing, A.O.B., M.V.G. and M.V.C.; supervision, M.V.C.; project administration, M.V.C.; funding acquisition, M.V.C. All authors have read and agreed to the published version of the manuscript.

**Funding:** "Coordenação de Aperfeiçoamento de Pessoal de Nível Superior (CAPES)", Finance Code 001 (student grants) and "Conselho Nacional de Desenvolvimento Científico e Tecnológico" (CNPq) (research grant 306902/2023-0).

**Conflicts of Interest:** The authors declare no conflict of interest.

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
