# Peer review of "Sanitizers Used for Fungal Spoilage Control in Dry-Fermented Cured Meat Production"

_fermentation, doi:10.3390/fermentation10030169_

Round 1

Reviewer 1 Report

Comments and Suggestions for Authors

Generally, the authors present a nice review. However, there are some problems in this manuscript. This paper may be considered for publication with some correction.
Please find my specific comments below:

1.      The authors reviewed several sanitizers used for fungal spoilage control; however, the relationship of sanitizers and dry-fermented cured meat was not well presented and discussed, so the title and the content of the manuscript were not matched well.

2.      Line 145-146, positive or negative electrical charges should be written at the upper corner of the atom.

3.      Line 159-160, ‘0.05 to 0.08% (600 to 800 parts per million)’ should be 0.05 to 0.08% (500 to 800 parts per million)’.

4.      Line 170-171, the concept of ‘organic matter’ was not clear.

5.      Line 186, ‘acid peracetic’ should be ‘peracetic acid’.

6.      Line 203, is it necessary to use two ‘pH’?

7.      Line 285, what was the meaning of ‘Yes’?

Author Response

Generally, the authors present a nice review. However, there are some problems in this manuscript. This paper may be considered for publication with some correction.
Please find my specific comments below:

  1. The authors reviewed several sanitizers used for fungal spoilage control; however, the relationship of sanitizers and dry-fermented cured meat was not well presented and discussed, so the title and the content of the manuscript were not matched well.
    1. We appreciate your comments and suggestions. Thank you for your time and effort. The term cured meat was corrected to dry-fermented meat (it was a translation mistake), so we believe the content is fitting better with the title. Unfortunately only a limited number of studies in this subject is available, so, sometimes we brought more broad information.
  2. Line 145-146, positive or negative electrical charges should be written at the upper corner of the atom.
    1. It was corrected.
  3. Line 159-160, ‘0.05 to 0.08% (600 to 800 parts per million)’ should be ‘0.05 to 0.08% (500 to 800 parts per million)’.
    1. Typo corrected.
  4. Line 170-171, the concept of ‘organic matter’ was not clear.
    1. The term was substituted by organic load, which is more used in literature. Additionally, the word soil was mentioned in L 141.
  5. Line 186, ‘acid peracetic’ should be ‘peracetic acid’.
    1.  
  6. Line 203, is it necessary to use two ‘pH’?
    1. Adjusted
  7. Line 285, what was the meaning of ‘Yes’?
    1. It was a typo error, we apologize.

Reviewer 2 Report

Comments and Suggestions for Authors

This work reviewed the disadvantages and factors that may interfere with the action of commonly used sanitizers by dry‑fermented cured meat industries, such as sodium hypochlorite, peracetic acid, and benzalkonium chloride. It seems to be suitable for publication in this journal. Comments:

1. The introduction should be re-written with improved logicality.

2. The previous review papers in this topic should be cited and discussed.

3. The framework and purpose of this review should be highlighted in Introduction.

4. What is the classification criterion of each sub-title?

5. The detection techniques and lifetimes of each sanitizer should be discussed.

Author Response

This work reviewed the disadvantages and factors that may interfere with the action of commonly used sanitizers by dry‑fermented cured meat industries, such as sodium hypochlorite, peracetic acid, and benzalkonium chloride. It seems to be suitable for publication in this journal.

We appreciate your comments and suggestions. Thank you for your time and effort.

Comments: 

  1. The introduction should be re-written with improved logicality.

The paragraph order was rearranged to meet the referee expectations.

  1. The previous review papers in this topic should be cited and discussed.

The 4 reviews available in the topics are cited (see below), but in my opinion the data from research studies are the ones that should receive more importance in a review paper.

Visconti V, Coton E, Rigalma K, Dantigny P: Effects of disinfectants on inactivation of mold spores relevant to the food industry: a review. Fungal Biology Reviews. 2021; v. 38(2):44-66. DOI: 10.1016/j.fbr.2021.09.004.

Bernardi, A. O.; Garcia, M. V.; Copetti, M. V. Food industry spoilage fungi control through facility sanitization. Current Opinion in Food Science. 2019, 29:28–34 . DOI: 10.1016/j.cofs.2019.07.006

Copetti, M. V. Sanitizers for controlling fungal spoilage in some food industries. Current Opinion in Food Science. 2023. v. 52, n. 101072. DOI:  https://doi.org/10.1016/j.cofs.2023.101072

Davies, C. R; Wohlgemuth, F.; Young, T.; Violet, J.; Dickinson, M.; Sanders, J. W.; et al. Challenges and evolving strategies for mold control in the food supply chain. Fungal Biology Reviews. 2021; v. 36:15–26. DOI: [http://dx.doi.org/10.1016/j.fbr.2021.01.003]

  1. The framework and purpose of this review should be highlighted in Introduction.

You can find this information from lines 102-111.

  1. What is the classification criterion of each sub-title?

Relevance.

  1. The detection techniques and lifetimes of each sanitizer should be discussed.

I apologize, but this is not in the scope of this review. This review is focused in applied microbiology, not chemistry.

Reviewer 3 Report

Comments and Suggestions for Authors

The authors described the effects of sanitizers, commonly used in the food industry for fungal control. Although the review provides significant and valuable information regarding the use of sanitizers and their efficiency, it can be improved.

 - The link between dry-fermented cured meat production, fungi spoilage, and sanitizer use should be better explained. It is unclear what surfaces are treated with sanitizers (e.g., work surfaces or food?) It should be explained in more detail for each of the investigated sanitizers.

- Please clearly indicate examples and data regarding the use and efficacy of each of the investigated sanitizers in the dry-fermented cured meat production

- Line 48. The authors should specify the ochratoxin A-producing species

- Line 54-58. Please correct the sentence and split it into two.

- Line 77- what species?

-Lines 123-127. Please add references

- Tables A1 and A2- please insert the relevant references in Tables

- Table A2. The text is missing in the last row, the third column. (Advantages- Low.)?

- Lines 285-286. Please explain the sentence: “It was then reported that washing with peracetic acid generated Yes, disinfection by-products, but in smaller quantities than with hypochlorite sodium”. What is “Yes”?

Comments on the Quality of English Language

- The manuscript needs extensive English editing.

Author Response

The authors described the effects of sanitizers, commonly used in the food industry for fungal control. Although the review provides significant and valuable information regarding the use of sanitizers and their efficiency, it can be improved.

We appreciate your comments and suggestions. Thank you for your time and effort.

 - The link between dry-fermented cured meat production, fungi spoilage, and sanitizer use should be better explained. It is unclear what surfaces are treated with sanitizers (e.g., work surfaces or food?) It should be explained in more detail for each of the investigated sanitizers.

Work surfaces. Clarification was done in L 62-75, L 155-157 (before start talking about each single sanitizer)

- Please clearly indicate examples and data regarding the use and efficacy of each of the investigated sanitizers in the dry-fermented cured meat production

This information can now be accessed in lines: L.190, L. 202-203, L. 277-279

- Line 48. The authors should specify the ochratoxin A-producing species

This information is now available in L41-44.

- Line 54-58. Please correct the sentence and split it into two.

The sentence was improved.

- Line 77- what species?

Information added (A. westerdijkiae, Penicillium polonicum, Aspergillus pseudoglaucus)

-Lines 123-127. Please add references

Added

- Tables A1 and A2- please insert the relevant references in Tables

Added

- Table A2. The text is missing in the last row, the third column. (Advantages- Low.)?

Low environmental impact, corrected

- Lines 285-286. Please explain the sentence: “It was then reported that washing with peracetic acid generated Yes, disinfection by-products, but in smaller quantities than with hypochlorite sodium”. What is “Yes”?

This was a typo error, we apologize. It was fixed.

Reviewer 4 Report

Comments and Suggestions for Authors

The authors describe the advantages and disadvantages of antimicrobials, as well as the influence of various factors on the antibacterial effect during the use of antimicrobials. This topic is important for the development of effective and safe sterilization techniques for food. The manuscript fits the theme of the journal. However, it still has some problems

1, please check the writing format and grammar errors

2. Toxicological data of supplementary fungicides and safety of current effective doses

3, at present, the combined use of fungicides may lead to synergistic effects, resulting in a decrease in the dosage of fungicides. Therefore, authors should discuss the effect of the combination of fungicides.

4. Please cite more literatures published in the last three years.

Comments on the Quality of English Language

Moderate editing of English language required

Author Response

The authors describe the advantages and disadvantages of antimicrobials, as well as the influence of various factors on the antibacterial effect during the use of antimicrobials. This topic is important for the development of effective and safe sterilization techniques for food. The manuscript fits the theme of the journal. However, it still has some problems

We appreciate your comments and suggestions. Thank you for your time and effort to improve this manuscript.

1, please check the writing format and grammar errors

A grammatical correction was carried out; hopefully, this new version will meet your expectations.

  1. Toxicological data of supplementary fungicides and safety of current effective doses

Unfortunately, this point is not the aim of this review. An important difference between fungicides and disinfecting is the contact time. These products are used for disinfecting work surfaces and the environment and should be rinsed after acting for about 15 min. They should be applied by experienced personnel using adequate protection. The maximum dosage allowed is defined by law.

3, at present, the combined use of fungicides may lead to synergistic effects, resulting in a decrease in the dosage of fungicides. Therefore, authors should discuss the effect of the combination of fungicides.

Currently, no data on the antifungal efficacy of combined or mixed sanitizers for use in food industries were found. The data available is limited, so we cannot provide such information. Actually, the opposite may also occur and this is a concern regarding sanitizers.

  1. Please cite more literatures published in the last three years.

Only a limited number of manuscripts relate to this revision's topic, and we tried to use them along with the manuscript.

Reviewer 5 Report

Comments and Suggestions for Authors

The fungal spoilage of food, cured meat in particular, is a subject of high concern. This means that the topic of this manuscript is quite important and actual. It is also well recognized that it is not an easy task to solve, and leads to health injuries and even victims.

However, a review must offer to the reader a systematic approach to the subject, with a deep discussion about causes and consequences of several alternatives, and a clear exploitation of literature data. The output must be a contribution to the understanding and practical information for the reader to take home, instead of a sum up of doubts and uncertainties.

The text reporting three sanitizers is quite limited. The use of bio sanitizers and the comparison with other alternatives, e.g. the use of radiation or pasteurization, would be appreciated. The use of sanitizers directly on the meat, or to disinfect the process equipment, or the work environment within the food industry plant should be analyzed separately.

The manuscript is more a disclosure text instead of a scientific paper. The comments are qualitative, with many generalities and missing of the scientific method in the analysis and utilization of the long citation list. Words or adjectives like low, little, high, slow, …. are not precise to be used in technical information. The sentences are, most of them, inconclusive, making the reader to put questions (how, when, why, how much, …) continuously. Some sentences must be even incomplete (e.g. line 102) or absolutely senseless (e. g. line 206), or including wrong words, such as “Yes” in line 285. Figure 1 is hard to understand and has no explanation. Tables are very useful to organize a great amount of information, but Table A2 is not a good example.

The correspondence author has too many published papers about the subject, and the current manuscript does not include innovation (or new interpretation), being reproduction. In the reference list, at least 13 are self-citations.

There is no add value to the reader with this generic text. The presence of organic matter as a hindrance to the antimicrobial action of sanitizers is very peculiar, as all foods are organic. Also the highlight of the importance of cleaning stages to reduce the fungal contamination seems to be quite obvious.

When using a disinfection plan (sanitizer type, dosage, application procedure, …) to a product, equipment or environment, it is useful to define a target or a reference strain, as well as a methodology to evaluate the efficiency of the disinfectant.

The trouble associated to the presence of spores, as well as the interference of several microorganisms in the same sample, makes the disinfection action still much trickier. All these questions should be included in the discussion of a thematic like the one presented in this manuscript.

Comments on the Quality of English Language

Minor corrections would be advantageous.

Author Response

The fungal spoilage of food, cured meat in particular, is a subject of high concern. This means that the topic of this manuscript is quite important and actual. It is also well recognized that it is not an easy task to solve, and leads to health injuries and even victims.

However, a review must offer to the reader a systematic approach to the subject, with a deep discussion about causes and consequences of several alternatives, and a clear exploitation of literature data. The output must be a contribution to the understanding and practical information for the reader to take home, instead of a sum up of doubts and uncertainties.

We appreciate your comments and suggestions, even if they were more of a critic's view than a referee's effort to improve a manuscript. This is not a systematic review, and we are sorry for not reaching your high expectations regarding this review. Perhaps this manuscript reflects the lack of quality data in the literature and the necessity of scientists to work also in basic applied microbiology and hygiene to solve some real and current problems faced by food industries.

The text reporting three sanitizers is quite limited. The use of bio sanitizers and the comparison with other alternatives, e.g. the use of radiation or pasteurization, would be appreciated. The use of sanitizers directly on the meat, or to disinfect the process equipment, or the work environment within the food industry plant should be analyzed separately.

This review is focused on chemical sanitizers that are allowed to be used in food industries. The number of compounds allowed is quite restricted, and the most used and reported active principles were considered in this review. Perhaps after the text improvement, you will realize that most of the literature available about the antifungal efficacy of such products comes from in vitro surface tests with no studies carried out on product surface or even on site. It is impossible to separate this information if the data is unavailable.

The manuscript is more a disclosure text instead of a scientific paper. The comments are qualitative, with many generalities and missing of the scientific method in the analysis and utilization of the long citation list. Words or adjectives like low, little, high, slow, …. are not precise to be used in technical information. The sentences are, most of them, inconclusive, making the reader to put questions (how, when, why, how much, …) continuously. Some sentences must be even incomplete (e.g. line 102) or absolutely senseless (e. g. line 206), or including wrong words, such as “Yes” in line 285. Figure 1 is hard to understand and has no explanation. Tables are very useful to organize a great amount of information, but Table A2 is not a good example.

Ok.

The correspondence author has too many published papers about the subject, and the current manuscript does not include innovation (or new interpretation), being reproduction. In the reference list, at least 13 are self-citations.

When someone is invited to write a review on a subject, perhaps it is because this author has a background on the topic. Most of the data available in the literature regarding the antifungal efficacy of sanitizers comes from my group experiments. Therefore, they are cited. If I cannot “self-cite” the few available data, no review manuscript can be done by myself. Check this review written by someone else (Effects of disinfectants on inactivation of mold spores relevant to the food industry: a review - ScienceDirect) and you will see similar literature used.

There is no add value to the reader with this generic text. The presence of organic matter as a hindrance to the antimicrobial action of sanitizers is very peculiar, as all foods are organic. Also the highlight of the importance of cleaning stages to reduce the fungal contamination seems to be quite obvious.

Even though it is “quite obvious” to academics, this adequate cleaning step is not being carried out by all food industries. Time is money, and mixed cleaning/disinfecting products, probably ineffective, are commercialized and intended to be applied in the sector. So, A + B=C formulae is still relevant nowadays, especially in open-access publications that can easily reach food industry personnel.  

When using a disinfection plan (sanitizer type, dosage, application procedure, …) to a product, equipment or environment, it is useful to define a target or a reference strain, as well as a methodology to evaluate the efficiency of the disinfectant.

Yes, we have the A. brasiliensis as standard strain for such tests however, no controlled tests in a food industry environment followinga disinfection plan were found. As mentioned in the text, the most accepted methodology for in vitro evaluation of sanitizer efficacy is the one from CEN.

The trouble associated to the presence of spores, as well as the interference of several microorganisms in the same sample, makes the disinfection action still much trickier.

It is hard to discuss without laboratory data to support these thoughts. For spores you mean ascospores? Or mixing fungi ordinary spores with bacteria spores?

All these questions should be included in the discussion of a thematic like the one presented in this manuscript.

Unfortunately, we cannot re-write this review, but I will be happy to see a such complete review published.

Round 2

Reviewer 2 Report

Comments and Suggestions for Authors

The authors did not adress my questions carefully.

Author Response

Dear Referee,

Could you be more specific about what was not addressed or justified in the previous round?

Reviewer 3 Report

Comments and Suggestions for Authors

The authors significantly improved their work.

Please add references in the following sentences. “These analyses may follow the methodology of the Association of Official Analytical Chemists (AOAC) or methods endorsed by the European Committee for Standardization (CEN) for liquid sanitizers (Ref ?). For smoke sanitizers, compliance with the French Standard (NF‑T‑72281) is essential, as it outlines the methodology for evaluating the effectiveness of smoke‑generating agents (Ref?).”

Author Response

References updated

Reviewer 4 Report

Comments and Suggestions for Authors

This manuscript has been revised according to comments. Its quality is improved. I suggest publishing it

Comments on the Quality of English Language

Minor editing of English language required

Author Response

Thank you very much for your time and effort improving the manuscript quality.

Reviewer 5 Report

Comments and Suggestions for Authors

The improvement efforts carried out by the authors are meritorious. The manuscript has been quite improved. The text is now much clearer, focused on the topic it addresses, easier to read and the statements are more based on literature support. The limitations of bibliographic sources to go deeper into certain aspects, as well as to refer more examples at working scale, are understandable and accepted.

-  line 8 of the 2nd paragraph of section 2.1.1 – in in vitro … (“in” is duplicated).

-       Second sentence before Conclusions:

««The presence of soil and water prevents the biodegradation of natural

environments due to their toxicity to many species of organisms’ aquatic and terrestrial

[100].»»

Maybe:

«« In the presence of soil and water they prevent the biodegradation of natural

environments due to their toxicity to many species of aquatic and terrestrial organisms [100]. »»

Author Response

Sorry for the mistakes.

Double "in" removed and the sentence adjusted to "The presence of quaternary ammonium compounds in soil and water prevents the biodegradation of natural environments due to their toxicity to many species of aquatic and terrestrial organisms "